# Evaluation of the Inhibitory Efficacy of Eugenol against the Pathogen of *Fusarium* Wilt in Ginger Seedlings

Xian Zhou [1,†], Hui-Hui Ma [1,†], Shi-Jie Xiong [1], Ling-Ling Zhang [1], Xue-Dong Zhu [2], Yong-Xing Zhu [1,*] and Li-Rong Zhou [1,*]

1 Spice Crops Research Institute, College of Horticulture and Gardening, Yangtze University, Jingzhou 434025, China; zhouxian20232023@163.com (X.Z.); 2021720843@yangtzeu.edu.cn (H.-H.M.); xsj19980418@163.com (S.-J.X.); llzhchina@163.com (L.-L.Z.)
2 Yudongnan Academy of Agricultural Sciences, Chongqing 408000, China; ozzy2311670@163.com
* Correspondence: xbnlzyx@163.com (Y.-X.Z.); zlr2021710830@163.com (L.-R.Z.)
† These authors contributed equally to this work.

**Abstract:** *Fusarium* wilt, mainly caused by *Fusarium oxysporum*, affects ginger yield and quality worldwide. To explore a safe and sustainable method of preventing and controlling *Fusarium* wilt, the inhibitory effects of eugenol on *F. oxysporum FOX-1* were analysed in this study. The results showed that eugenol inhibited the reproductive and vegetative growth of *F. oxysporum FOX-1* in vitro. Microscopic observations revealed that eugenol disrupted the hyphal morphology of *F. oxysporum*. In addition, eugenol destroyed the cell membrane integrity of the pathogenic fungi, resulting in the leakage of *F. oxysporum FOX-1* intercellular contents, including electrolytes, soluble proteins, nucleic acids, and malonydialdehyde. Results of an in vivo inoculation test showed that eugenol exerted a strong inhibitory effect on *Fusarium* wilt in ginger seedlings. In summary, eugenol had an inhibitory effect on the growth of *F. oxysporum FOX-1* and controlled *Fusarium* wilt in ginger seedlings. These findings provide a foundation for future development of botanical antifungal agents to manage *Fusarium* wilt.

**Keywords:** *Fusarium* wilt; eugenol; essential oil; antifungal activity; *Zingiber officinale*





## 1. Introduction

Ginger (*Zingiber officinale* Roscoe) is a perennial herb belonging to the family Zingiberaceae. It has been widely used as a spice and condiment, as well as in traditional medicine because of its antibacterial, anti-inflammatory, antipyretic, antioxidant, and diuretic properties [1,2]. In recent years, the supply and demand for ginger have increased, but the annual output is limited by viruses, bacteria, and fungi [3]. *Fusarium oxysporum* is a common soil-borne pathogen that causes wilt and root rot in many crops, including ginger [4]. This condition is often referred to as *Fusarium* wilt, and it generally leads to a yield loss of approximately 50%, reaching more than 80% in some areas [5]. Chemical fungicides have been widely used in the commercial production of ginger to control *Fusarium* wilt. However, using pesticides causes safety problems such as pesticide residues, environmental pollution, and pathogen resistance [6,7]. According to the Ministry of Agriculture and Rural Affairs in China, if the national pesticide utilisation rate increases by 1%, the use of pesticides is reduced by nearly 30,000 tons, equivalent to a reduction of 1.7 billion yuan in farmers' production inputs, which is also conducive to reducing pesticide residues and protecting the ecological environment [8]. Therefore, it is important to explore and utilise efficient, new, and environmentally friendly pesticide treatments.

Plant extracts are among the most widely studied substitutes for chemical fungicides because of their diverse functions [9]. Eugenol (4-allyl-2-methoxyphenol), a monoterpene, exists in the essential oils of many kinds of plants, has various reported medicinal properties, and is widely used in the medical field [10,11]. Recently, the antimicrobial effects of eugenol

against various pathogenic microorganisms have been reported. Wang et al. [12,13] found that eugenol, as the main active antibacterial ingredient of clove essential oil, exerted a strong inhibitory effect on *Fusarium solani* and *Fusarium graminearum*. However, whether eugenol exerts antimicrobial effects against *Fusarium* wilt in ginger seedlings has not been reported.

This study aimed to (1) study the effects of eugenol on the reproductive and vegetative growth of *F. oxysporum FOX-1* in vitro, (2) explore the mechanism of action of eugenol using microscopic observations and cell membrane integrity detection, and (3) evaluate the inhibitory efficacy of eugenol against *Fusarium* wilt in ginger seedlings. Our results will help to provide an efficient alternative approach to control *Fusarium* wilt.

## 2. Materials and Methods

### 2.1. Preparation of Fungal Pathogen

The *F. oxysporum* strain (FOX-1) was isolated by the Institute of Special Plants, Chongqing University of Arts and Sciences, and it was identified by Prof. Yi-Qing Liu [14]. The *F. oxysporum FOX-1* was cultured in Potato Dextrose Agar (PDA: potato 200 g/L, glucose 20 g/L, and agar 18 g/L) and incubated in the dark at 28 °C for 5 days, and each plate was eluted with 5 mL of sterilised ultrapure water and filtered through two layers of swabs and microscope paper. The samples were then counted by a hematocyte counter, and the concentration of spore suspension was adjusted to $1 \times 10^6$ spores/mL.

### 2.2. In Vitro Inhibitory Activity of Eugenol against F. oxysporum FOX-1

The inhibitory effect of eugenol on the mycelial growth of *F. oxysporum FOX-1* was tested according to the method published by Wang et al., with some modifications [15]. Eugenol (CAS-No. 93-53-0; purity 98%) was purchased from Aladdin Chemical Reagent Co., Ltd. (Shanghai, China). An amount of 3.752 mL of eugenol was aspirated and dissolved with 30% ethanol ($v/v$) to configure 4 g/L of eugenol reserve solution. The configured reserve solution was added into the PDA medium (at about 50 °C) to configure the drug-containing medium with mass concentrations of 0.125, 0.25, 0.5, 1, and 2 g/L, respectively, and the plates were poured after shaking well. Then, 8 g/L of chlorothalonil (Chtl.) was used as the positive drug control [16]. A medium containing 0 g/L of eugenol essential oil and 30% ethanol ($v/v$) was used as the negative control, and it was centrally inoculated with d = 6 mm cultured mycelium of *F. oxysporum FOX-1*. Subsequently, 6 mm mycelial discs of *F. oxysporum FOX-1* were inoculated at the centre of the medium and incubated at 28 °C for 5 d. The colony diameter was measured every 24 h by using the crossover method. The colony diameter of the *F. oxysporum* was measured every 24 h after inoculation. The concentration of eugenol on the plate exhibiting no hyphal growth within 48 h was deemed the minimum inhibitory concentration (MIC) [17]. The growth inhibition rate was calculated, and the experimental design included three replicates for each treatment group. The logarithmic concentration–inhibition rate probability value method was used to find the regression equation of virulence and the half-effect concentration ($EC_{50}$).

In 1 mL ($1 \times 10^6$ spores/mL) of *F. oxysporum FOX-1* spore suspension, different volumes of eugenol were added to the suspension to achieve final concentrations of 0, $\frac{1}{2}$MIC, and MIC. Then, 100 μL of the mixture was added dropwise to the centre of a concave slide. The slides were placed in sterile petri dishes with wet filter paper and incubated in an incubator at 28 °C. After 12 h, a microscopic observation was made, and the spore germination number was counted if the germination tube was longer than half of the spore diameter. Three repetitions of each treatment were made, three fields of vision were examined in each repetition; more than 100 spores were examined each time, and the germination rate of the spores was calculated.

*2.3. Antifungal Mechanisms of Eugenol against F. oxysporum FOX-1*

2.3.1. Scanning Electron Microscope (SEM) Observation for Mycelia

A suspension of $1 \times 10^4$ spores/mL of *F. oxysporum FOX-1* was added to 50 mL of Potato Dextrose Broth (PDB: potato 200 g/L, glucose 20 g/L) medium containing eugenol $(0, \frac{1}{2}\text{MIC}, \text{and MIC})$, and incubated for 3 d at 28 °C and 150 rpm on a constant temperature shaker (MaxQ HP 436, Shanghai, China). The mycelium was collected by filtration through eight layers of sterile gauze, rinsed three times with 0.1 mol/L phosphate buffer solution (PBS, pH 7.2), resuspended in PBS (pH 7.2) three times, and resuspended in PBS. Chlorothalonil at 8 g/L was added as a positive control, and 0 g/L of eugenol essential oil and 30% ethanol were added as a negative control. The samples were prepared according to Yu et al. [18], and each treatment was fixed with 2.5% glutaraldehyde fixative and placed in a refrigerator at 4 °C to freeze-cure for 12 h. The mycelium of each treatment was rinsed three times with PBS (pH 7.2) and dehydrated using volume fractions of 30%, 50%, 70%, and 90% ethanol solutions and anhydrous ethanol, each time for 20 min. The samples were fixed and dried using a vacuum freeze dryer (FD-1A-50, Shanghai Bilon Instrument Co., Ltd., Shanghai, China). The samples were then sputter-coated using an ion sputter coating machine (SC7620, Quorum Technologies Ltd., East Sussex, UK) for 120 s. Finally, the changes in the surface morphology of the mycelium were observed using SEM (JEOL JSM-6390LV, Beijing, China) at a magnification of 5000×. Each treatment was measured five times, and each measurement comprised 10–20 microscope observations. The experiment was repeated three times.

2.3.2. Assay of Cell Membrane Integrity

Bacterial cakes (6 mm in diameter) were punched from the edge of the *F. oxysporum FOX-1* plates, which had been activated for 5 d, inoculated into PDB medium containing different effective concentrations of eugenol essential oils $(0, \frac{1}{2} \text{MIC}, \text{and MIC})$, and incubated for 12 h at 28 °C on a shaker with a constant temperature of 180 rpm, then centrifuged (Centrifug, Sorvall ST8, Shanghai Bilon Instrument Co. Ltd., Shanghai, China) for 5 min at 5000× $g$ at 4 °C to collect the hyphae, which were washed three times in PBS and resuspended in 5 mL of PBS. The samples were stained with 10 μg/mL of PI (Solarbio Life Science, Beijing, China) and incubated in the dark at 30 °C for 20 min. After staining, the samples were rinsed three times with PBS to remove the residual dye. Finally, the samples were observed by laser confocal microscopy (Thermo Fisher Scientific, Waltham, MA, USA) (excitation: 545 nm; emission: 590 nm), three replicates were set up for each treatment replicate, three fields of view were observed randomly each time, and the whole experiment was repeated three times.

2.3.3. Detection of Cellular Leakage

Three grams of mycelium (fresh weight) were weighed and suspended in PBS (30 mL) containing different concentrations of eugenol $(0, \frac{1}{2}\text{MIC}, \text{and MIC})$. The treatments were shaken at 28 °C, 180 r/min on a constant temperature shaker, and after incubation for 0, 2, 3, 4, 6, 8, 9, 10, and 12 h, they were centrifuged at 10,000× $g$ at 4 °C for 10 min to obtain the supernatant.

The relative conductivity was calculated using Equation (1):

$$\text{The relative conductivity (\%)} = (L1 - L0)/(L2 - L0) \times 100. \tag{1}$$

where $L1$ is the conductivity of the culture solution measured at a given incubation time, $L0$ is the conductivity of the culture measured at the beginning of the incubation, and $L2$ is the conductivity measured after inactivation of the mycelium [19].

The mycelia were treated according to the above method, and the absorbance of the supernatant was measured using an ultraviolet spectrophotometer (UV-2800A, Mapada, Shanghai, China) at 260 and 280 nm to determine the nucleic acid and protein concentrations. The malonydialdehyde (MDA) content of *F. oxysporum* mycelia was determined

using an MDA content assay kit (Nanjing Jiancheng Bioengineering Institute, Nanjing, China). The experimental design consisted of three replicates for each treatment, and the experiment was repeated three times.

### 2.3.4. Detection of Ergosterol Content in the Cytoplasmic Membrane

The ergosterol content in the cytoplasmic membrane of *F. oxysporum FOX-1* mycelia was determined, using a method previously described by Wang et al. [20]. Briefly, mycelia of *F. oxysporum FOX-1* were added to PDB containing different concentrations of eugenol (0, $\frac{1}{2}$ MIC, and MIC) and co-cultured for 2 d in darkness at 28 °C. The cultured control groups were treated in a similar manner but without eugenol. After incubation, the mycelium was collected by filtration for each group of samples and washed three times with PBS. The net wet weight of the mycelia was recorded. To extract ergosterol, 5 mL freshly prepared 25% (*w/v*) ethanol–KOH solution was added to the sample, after which the sample was shaken for 2 min and incubated at 85 °C for 4 h. After saponification, 5 mL of heptane and 2 mL of sterile distilled water were added, and the mixtures were vigorously shaken for 15 min before the layers were separated at 25 °C for 1 h. The absorbance of the heptane layer was measured using an ultraviolet-visible spectrophotometer (Thermo Fisher Scientific, Inc., Waltham, MA, USA). The ergosterol and the sterol intermediate 24(28)-dehydroergosterol generated characteristic peaks at 282 and 230 nm, respectively. The ergosterol content was calculated as a percentage of the wet weight of the cells using the following Equation (2):

$$\text{Ergosterol (\%)} = A_{282}/(\text{mycelia weight} \times 290) - A_{230}/(\text{mycelia weight} \times 518) \quad (2)$$

where 290 and 518 are the E values (in percentages per centimetre) determined for the crystalline ergosterol and 24(28)-dehydroergosterol, respectively, and the mycelial weight is the net wet weight (g). The experimental design consisted of three replicates for each treatment, and the experiment was repeated three times.

### 2.4. *In Vivo Inhibitory Efficacy of Eugenol on F. oxysporum FOX-1 Infection*

Healthy bamboo root ginger seedlings (grown for 2 months) with uniform growth were selected as inoculation materials, and the seedlings were divided into 3 groups: the negative control group (inoculated with sterile water), positive control group (inoculated with pathogenic bacteria only and positive drug control 8 g/L of Chtl.), and essential oil treatment group (pathogenic bacteria + plant essential oil). A syringe was used to inject 1 mL of *Fusarium oxysporum* spore suspension ($1 \times 10^6$ spores mL$^{-1}$) into the rhizomes of the seedlings. After 24 h, eugenol solutions at concentrations of 0, $\frac{1}{2}$MIC, 1MIC, 2MIC, and 20 mL were uniformly sprayed on the pot-planted seedlings, and 15 plants were planted in each group of treatments, which were incubated in a greenhouse at a temperature of 30 °C, relative humidity of 75–85%, and a light period of 12 h. The ginger seedlings were incubated in a greenhouse at 30 °C, 75–85% relative humidity, 12 h of light, and under normal care and management. According to the method described by Pan et al. [21], the ginger seedlings were graded for *Fusarium* wilt after 15 days. The disease incidence, disease index, and control effect were calculated as follows:

$$\text{Disease incidence (\%)} = \text{Number of diseased plants}/\text{total number of plants} \times 100 \quad (3)$$

$$\text{Disease index} = [\Sigma \, (\text{Number of plants in each class} \times \text{class value in each class})/(\text{number of observed plants} \times \text{highest value in the evaluation scale})] \times 100 \quad (4)$$

$$\text{Control efficacy (\%)} = (\text{Disease index of the control} - \text{disease index of the treatment})/\text{disease index of the control} \times 100 \quad (5)$$

*2.5. Statistical Analyses*

All the experiments were repeated at least three times, and the corresponding values are shown as mean ± standard error. All the data were analysed using SPSS software (v14.0, SPSS Inc., Chicago, IL, USA). Duncan's new complex polarization method was used for significance testing ($p < 0.05$).

## 3. Results

*3.1. Eugenol Suppressed Mycelial Growth and Spore Germination of F. oxysporum FOX-1*

Different concentrations of eugenol showed different degrees of inhibition of *F. oxysporum FOX-1* mycelial growth (Figure 1A). The colony diameter of CK was 5.97 ± 012 cm after 5 d of incubation, and the colony diameters of the eugenol treatments ranged from 0.60 ± 0.00 to 4.95 ± 0.25 cm (Table 1). We found that 0.5 g/L of eugenol maintained a high inhibitory effect on the growth of *F. oxysporum FOX-1* mycelium throughout the whole experimental process, and the mycelium was completely unable to grow when the concentration of eugenol was 1 g/L. Meanwhile, 8 g/L of chlorothalonil also completely inhibited mycelial growth, with no significant difference from 1 g/L of eugenol. To eliminate the effect of ethanol, we studied the effect of 30% ethanol on the colony growth of *F. oxysporum FOX-1* and found that it had little effect on the colony growth ($p > 0.05$; Figure 1A, Table 1).

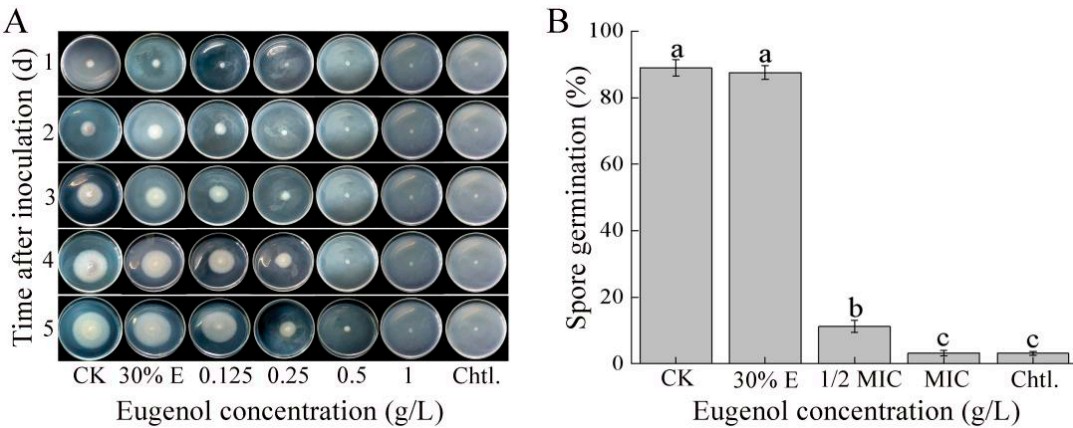

**Figure 1.** Effects of different concentrations of eugenol on *F. oxysporum FOX-1* growth. (**A**) Colony morphology after 5 d of treatment. (**B**) Spore germination rate after 12 h of treatment. The bars on the columns represent standard deviations, and different letters above the bars represent significant differences ($p < 0.05$). 30% E, 30% ethanol; CK, control (0 g/L of eugenol); Chtl, chlorothalonil (8 g/L).

**Table 1.** Eugenol suppressed *F. oxysporum FOX-1* growth in vitro.

| Concentration | Colony Diameter | Inhibition Rate | Virulence Equation | EC$_{50}$ | R$^2$ | MIC |
|---|---|---|---|---|---|---|
| (g/L) | (cm) | (%) | | (g/L) | | (g/L) |
| 0 | 5.97 ± 012 [a] | 0.00 ± 0.00 [e] | | | | |
| 30% ethanol | 5.94 ± 0.32 [a] | 0.00 ± 0.00 [e] | | | | |
| 0.125 | 4.95 ± 0.25 [b] | 18.99 ± 4.59 [d] | | | | |
| 0.25 | 2.57 ± 0.47 [c] | 63.38 ± 8.69 [c] | y = 18.26x + 47.879 | 0.235 | 0.3728 | 0.5 |
| 0.5 | 0.78 ± 0.18 [d] | 96.59 ± 3.41 [b] | | | | |
| 1 | 0.60 ± 0.00 [e] | 100.00 ± 0.00 [a] | | | | |
| 8 (Chtl.) | 0.60 ± 0.00 [e] | 100.00 ± 0.00 [a] | | | | |

Different superscript letters over the columns show significant differences ($p < 0.05$). Chtl, chlorothalonil (8 g/L).

Furthermore, eugenol significantly suppressed the spore germination of *F. oxysporum FOX-1* (Figure 1B). Eugenol treatment at $\frac{1}{2}$MIC decreased the spore germination of *F. oxysporum FOX-1* by 90.22%, whereas the spore germination of *F. oxysporum FOX-1* was

almost completely inhibited at the MIC of eugenol, and there was no significant difference between 0.5 g/L of eugenol and 8 g/L of chlorothalonil ($p > 0.05$; Figure 1B).

### 3.2. Eugenol Damages the Mycelial Morphology of F. oxysporum FOX-1

Figure 2 presents the image of mycelial changes obtained via SEM after 3 d of eugenol treatment. In the CK group, the mycelial morphology of *F. oxysporum* was normal, the surface was smooth, and the thickness was uniform (Figure 2A). With the increase in eugenol concentration, the mycelium is usually wrinkled and rough; correspondingly, the structural integrity of the cells was damaged to varying degrees, especially in the MIC group (Figure 2B,C).

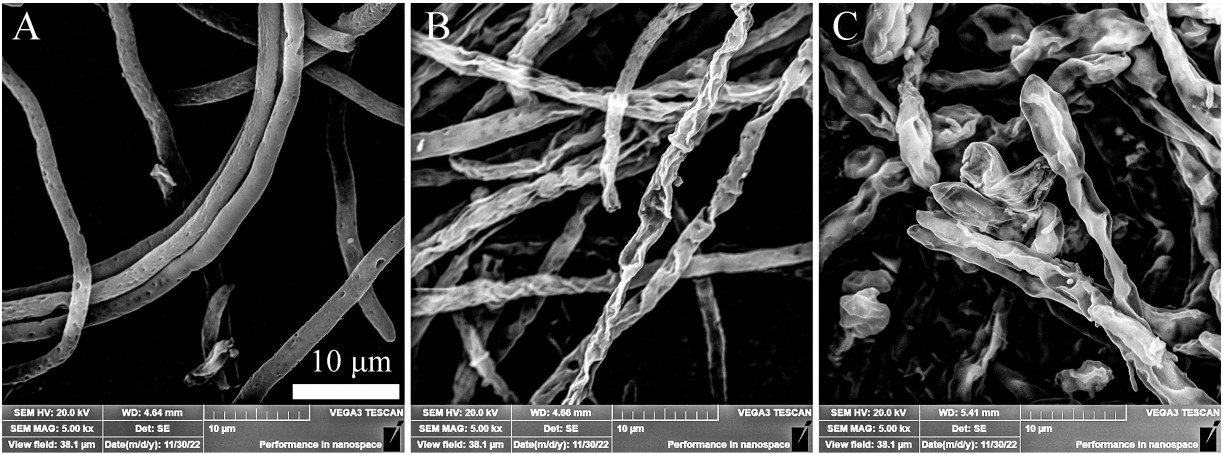

**Figure 2.** SEM images of *F. oxysporum FOX-1* mycelia. (**A**) CK, treatment with 0 g/L of eugenol. (**B**) $\frac{1}{2}$MIC, treatment with 0.25 g/L of eugenol. (**C**) MIC, treatment with 0.5 g/L of eugenol. SEM HV:20.0 kv; SEM MAG: 5.00 kv.

### 3.3. Eugenol Damaged the Cell Membrane of F. oxysporum FOX-1

As shown in Figure 3A, at 2 h and 12 h after treatment, the relative conductivity continuously increased with the increase in eugenol concentration. After 12 h treatment, the relative conductivity of *F. oxysporum FOX-1* treated with $\frac{1}{2}$ MIC and MIC was significantly increased by 2.86- and 3.84-fold, respectively ($p < 0.05$; Figure 3A), compared to that of CK. These results suggest that eugenol destroys the cell membrane of *F. oxysporum*, leading to leakage of intracellular electrolytes and increased relative conductivity.

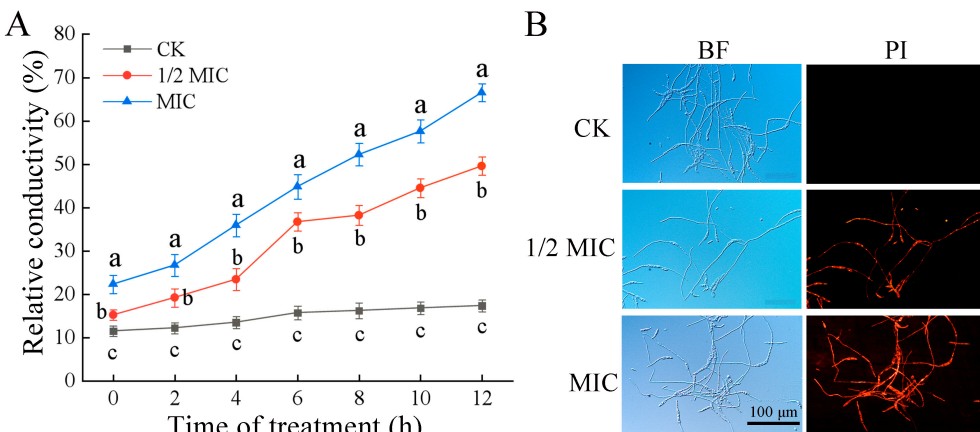

**Figure 3.** Eugenol affected the cell membrane permeability and integrity of *F. oxysporum FOX-1*. (**A**) Relative conductivity of mycelia and (**B**) PI staining observation of mycelia. BF, bright field; PI, PI staining. The bars on the curves represent standard deviations, and different letters at the same time point represent significant differences ($p < 0.05$).

As shown in Figure 3B, the *F. oxysporum FOX-1* mycelia in the CK showed no red fluorescence, indicating that they were intact. After 2 d of treatment with $\frac{1}{2}$MIC, the mycelia of *F. oxysporum FOX-1* maintained cell membrane integrity. However, most of the mycelia lost cell membrane integrity after the MIC treatment. Based on these results, we concluded that eugenol destroys the cell membrane of *F. oxysporum FOX-1*.

### 3.4. Eugenol Induced Cellular Leakage of F. oxysporum FOX-1

Figure 4A,B show that the soluble protein and nucleic acid leakage were almost unchanged in the control group. After eugenol treatment, the mycelial leakage of *F. oxysporum FOX-1* significantly increased compared to that in the control. Soluble protein and nucleic acid leakage were observed after 2 h of eugenol treatment, and the leakage continued for more than 8 h. Compared to the control group, the soluble protein and nucleic acid leakages increased after 12 h of eugenol treatment at the MIC by 2.45- and 6.58-fold, respectively.

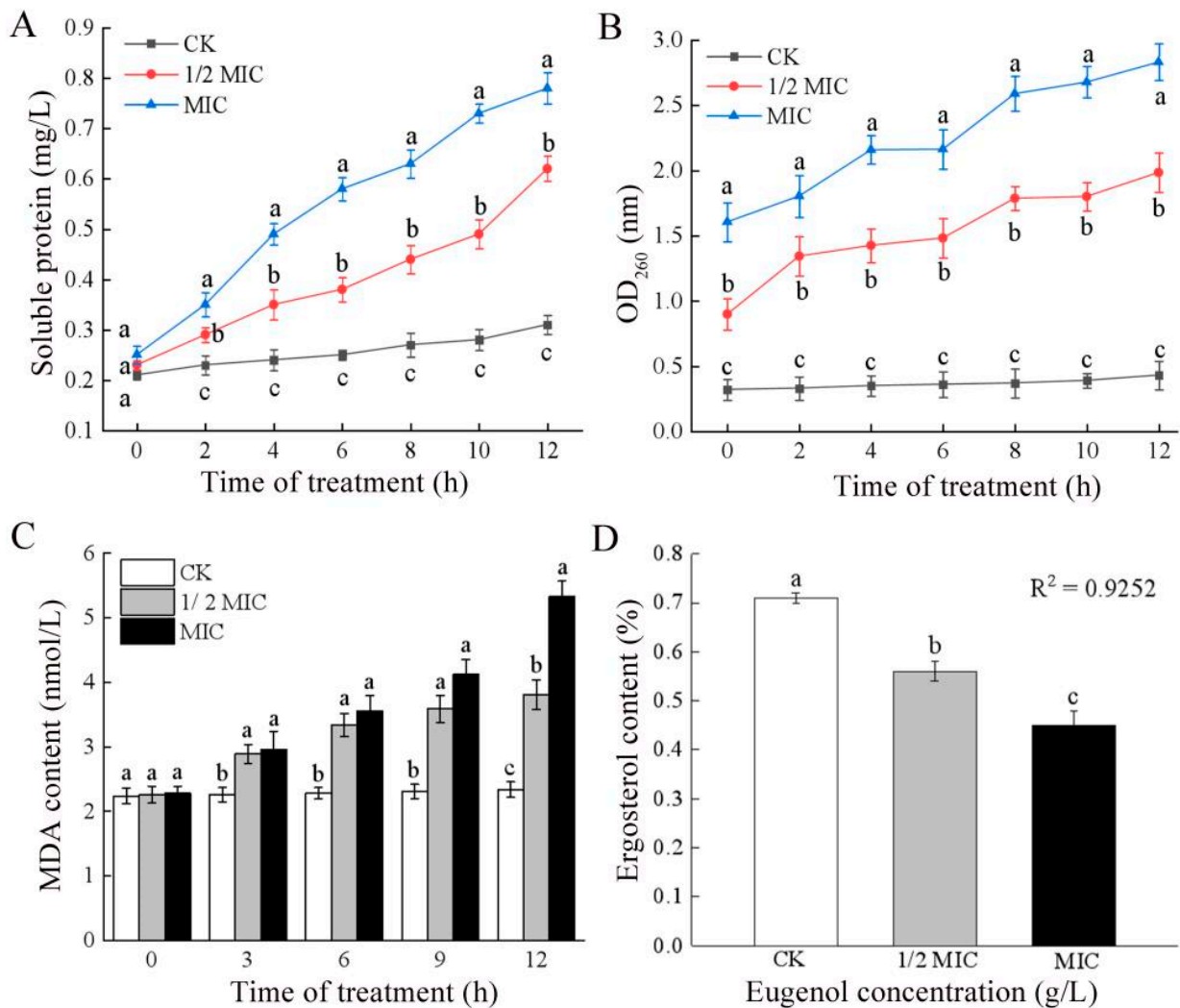

**Figure 4.** Effects of different concentrations of eugenol on (**A**) soluble protein, (**B**) nucleic acids, (**C**) MDA, and (**D**) ergosterol contents in *F. oxysporum FOX-1*. The bars on the curves represent standard deviations, and different letters at the same time point represent significant differences ($p < 0.05$).

### 3.5. Eugenol Inhibition of Ergosterol Synthesis

As shown in Figure 4C,D, compared to the CK, the $\frac{1}{2}$MIC and MIC treatment groups showed 1.01- and 2.07-fold increases in the MDA content in *F. oxysporum FOX-1*, respectively.

Compared to the CK group, 21.59% and 36.15% decreases in ergosterol content were observed in *F. oxysporum FOX-1* in the 1/2 MIC and MIC treatment groups, respectively.

### 3.6. Eugenol Controlled Fusarium Wilt of Ginger Seedlings

Pot experiments were performed to further clarify the effects of eugenol on *F. oxysporum*. As shown in Table 2, 15 d after *F. oxysporum FOX-1* inoculation, the control effects of the 2 g/L of eugenol and 8 g/L of chlorothalonil were 47.11% ± 1.51% and 45.48% ± 2.40%, respectively. Figure 5 shows the efficiency of eugenol in controlling *F. oxysporum* after inoculation for 15 d. Compared with the pathogen control plants, the signs of plant *Fusarium* wilt were significantly reduced.

**Table 2.** Inhibition effects of eugenol against *Fusarium* wilt of ginger seedlings.

| Groups | Treatment | Eugenol Concentration (g/L) | Disease Incidence (%) | Disease Index(%) | Control Efficacy (%) |
|---|---|---|---|---|---|
| Negative control | water | 0 | - | - | - |
| Pathogen control | *F. oxysporum* | 0 | 100.00 ± 0.00 a | 68.1 ± 0.81 a | 0.00 ± 0.00 d |
| Positive control | *F. oxysporum* + Chtl. | 0 | 25.33 ± 1.45 d | 36.00 ± 0.87 d | 47.11 ± 1.51 a |
| | *F. oxysporum* + eugenol | 0.5 | 66.33 ± 1.20 b | 55.84 ± 1.93 b | 18.00 ± 2.70 c |
| Treatment | *F. oxysporum* + eugenol | 1 | 45.33 ± 1.76 c | 44.97 ± 1.24 c | 33.90 ± 2.60 b |
| | *F. oxysporum* + eugenol | 2 | 25.67 ± 1.45 d | 37.09 ± 1.18 d | 45.48 ± 2.40 a |

Different letters over the columns show significant differences ($p < 0.05$).

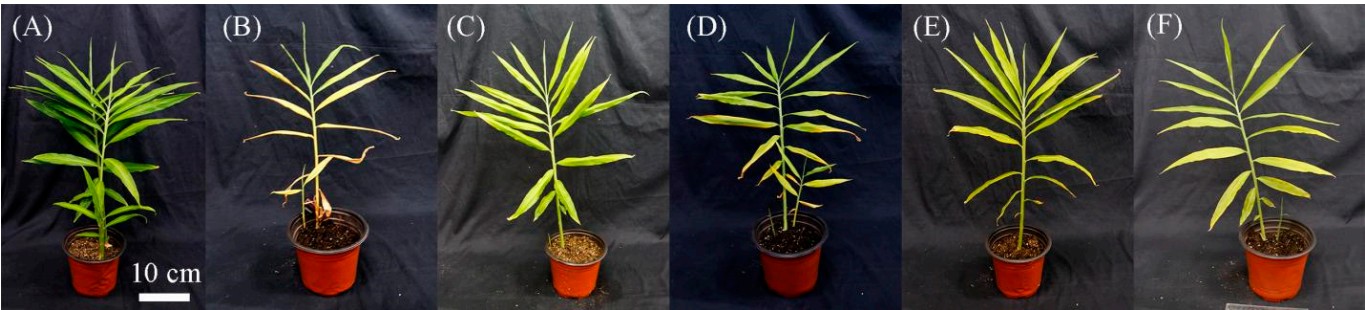

**Figure 5.** Effect of (**A**) negative control (sterile water), (**B**) pathogen control, (**C**) positive control, chlorothalonil (8 g/L), (**D**) 0.5 g/L of eugenol, (**E**) 1 g/L of eugenol, and (**F**) 2 g/L of eugenol on *F. oxysporum* inoculated ginger seedlings after 15 days of treatment.

## 4. Discussion

*Fusarium oxysporum* is a pathogen that causes *Fusarium* wilt in ginger and is one of the main factors limiting ginger production [22]. Owing to the limitations of pesticides, the use of plant metabolites to control *F. oxysporum* has attracted attention in research. The results of this experiment showed that eugenol treatment significantly inhibited *F. oxysporum FOX-1* mycelial growth and spore germination. The EC$_{50}$ value of eugenol against *F. oxysporum FOX-1* was 0.235 mg/mL, which is lower than that of some other previously reported plant-derived fungicides like *Lavendula angustifolia* (37.2 mg/mL) and essential oils from *Cinnamomum cassia* (50 mg/mL) [23,24], suggesting that eugenol has a favourable antifungal effect against *F. oxysporum* and that it shows promise as a botanical fungicide. A previous study showed that 52.77- and 166.37-µg/mL of eugenol treatment damaged the cell walls of fungal cells infecting *Rhizoctonia solani* and resulting in cell shrinkage [25]. Here, the SEM results showed that eugenol destroyed the hypha morphology of *F. oxysporum*, and the mycelium surfaces showed folds and obvious morphological changes at the MIC of eugenol. Similarly, Bai et al. reported that eugenol treatment results in obvious changes in the morphology of *Shigella flexneri* [26].

As the primary barrier for cells, the cell membrane has an important role in material, information, and energy exchange and supports normal metabolism and homeostasis [27]. The lipophilicity of eugenol allows it to enter between the fatty acid chains of the membrane

lipid bilayers, altering their fluidity and permeability [28]. Propidium iodide cannot penetrate the cell membrane and is usually removed from living cells [29]. In the present study, more cells in the eugenol-treated group were stained with PI than those in the CK group, indicating that a significant percentage of the cells lost membrane integrity. Similarly, Ju et al. found that extracellular PI penetrates and integrates into the DNA of citral- and eugenol-treated *Aspergillus niger* cells, indicating that these compounds damage the *Aspergillus niger* cell membrane [29]. In the current study, we detected comparably conspicuous leakage of the cellular contents of *F. oxysporum FOX-1* Specifically, the relative conductivity and nucleic acid and extracellular protein concentrations showed an obvious increasing trend with increasing eugenol treatment concentrations and treatment time (Figures 3B and 4A,B). Similarly, Sun et al. [30] found that eugenol damages the integrity of cell membranes, promoting the dissolution and leakage of macromolecules such as nucleic acids and proteins. These results indicate that the antifungal mechanism of eugenol may be attributed to its ability to destroy the integrity of the fungal cell membrane, leading to disruption of normal physiological metabolism and eventual cell death.

MDA content is an indicator of membrane lipid peroxidation in cells, reflecting the degree of peroxidation and cell membrane damage [31]. Notably, ergosterol is a major sterol component of the fungal cell membrane and has an important role in maintaining the integrity of the cell membrane; a decrease in ergosterol content could lead to the destruction of the cell membrane [26,32]. Many studies have shown that fungicides take on an inhibitory role by destroying the integrity of the cell membrane [33–35]. Zhao et al. found that the MDA content of *Rhizoctonia solani* sharply increased, and the ergosterol content was significantly reduced in the eugenol group. These results further confirm that the antifungal mechanism of eugenol is primarily associated with the destruction of membrane integrity [25].

Similarly, in the current study, eugenol treatment significantly reduced the ergosterol content of fungal cells, suggesting that eugenol destroyed the cell membrane by inhibiting the ergosterol synthesis. In addition, eugenol treatment increased the MDA contents, indicating that increased lipid peroxidation occurred in *F. oxysporum FOX-1*, which aggravated cell membrane damage and inhibited mycelial growth.

When assessing the suitability of natural products for use as commercial fungicides, both in vivo and in vitro experiments are crucial to ensure the products have the necessary efficacy [36]. In this study, in vitro studies of eugenol supported its potential as an antifungal agent against *F. oxysporum FOX-1*. Thus, we further investigated its in vivo control effect on *Fusarium* wilt, finding that it showed good efficacy in controlling *Fusarium* wilt in ginger seedlings and had a control efficacy similar to that of chlorothalonil. Considering that eugenol is safe, has low toxicity and drug resistance, and easily decomposes without residues as a natural product, we recommend its use to manage *Fusarium* wilt in the future. However, to further improve its antifungal efficacy against ginger *Fusarium* wilt, further studies should explore its mechanism of action on ginger *Fusarium* wilt at the molecular level, understand its targets, and provide a scientific basis for field control.

## 5. Conclusions

In this study, the soil-borne pathogen *F. oxysporum FOX-1* was the target, and it was found that eugenol inhibited the growth of the *F. oxysporum FOX-1* mycelium and spore germination, and its inhibitory effect had a concentration-dependent characteristic. Further study of its mechanism of action from the perspective of the cell membrane found that eugenol acted on the cell membrane of *Fusarium spinosum*, damaged the cell structure of the pathogenic fungi, and destroyed the integrity of the cell membrane and its permeability. Owing to the intracellular material spillover, the fungi cannot maintain normal physiological activities to achieve an antimicrobial effect. Finally, through potting experiments on bamboo root ginger for prevention and efficacy measurement, it was found that there was no significant difference between the control effect of treatment with 2 g/L of eugenol and 8 g/L of chlorothalonil. The results of this study can be used to provide a theoretical

basis and promote the development of highly efficient, plant-derived, and environmentally friendly agents for the control of soil-borne diseases caused by *Fusarium* species.

**Author Contributions:** Conception and design, L.-R.Z.; conducting experiments and writing—original draft preparation, X.Z., H.-H.M., and S.-J.X.; writing—review and editing, Y.-X.Z. and X.-D.Z.; supervision, Y.-X.Z.; data analysis, L.-L.Z. All authors have read and agreed to the published version of the manuscript.

**Funding:** This research was funded by the Key R&D Projects in Hubei Province, China (2022BBA0061) and the Key R&D Projects in Hubei Province, China (2021BBA096).

**Institutional Review Board Statement:** Not applicable.

**Informed Consent Statement:** Not applicable.

**Data Availability Statement:** Not applicable.

**Conflicts of Interest:** The authors declare no conflict of interest.

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
