# Peer review of "Evaluation of the Inhibitory Efficacy of Eugenol against the Pathogen of Fusarium Wilt in Ginger Seedlings"

_horticulturae, doi:10.3390/horticulturae9091024_

Round 1

Reviewer 1 Report

1. Title should be in coherent with the subject you are investigating, the causal organism- and the host. You may consider to put in the title

2. Isolates need over-all confirmatory test by molecular characterization, any NCBI code?

3. Completely describe methods properly (see my suggestions).

4. Data presentation is ok but needs to re-arrange - see my comments and suggestions

5. Conclusion should be an over results of the study based on the target objectives.

6. Recast the discussions following the sequence of presentation of data.

Needs minor editing

Author Response

Responses to reviewer 1
Title: Evaluation of the inhibitory efficacy of eugenol against the pathogen of Fusarium wilt in ginger seedlings.

Dear reviewer:

We gratefully thank you time spend making constructive remarks and useful suggestions, which has significantly raised the quality of the manuscript and has enable us to improve the manuscript.

Q1:Title should be in coherent with the subject you are investigating, the causal organism- and the host. You may consider to put in the title.

Rs: Thanks for your suggestion. The title has been changed to “Evaluation of the inhibitory efficacy of eugenol against the pathogen of Fusarium wilt in ginger seedlings

Q2:Isolates need over-all confirmatory test by molecular characterization, any NCBI code?

Rs: Thanks. “The F. oxysporum strain (FOX-1) was isolated by the Institute of Special Plants, Chongqing University of Arts and Sciences, and it was identified by Prof. Yi-Qing Liu, they didn't upload NCBI code. However, there are published articles proving that。

  • Liu, Y., Wisniewski, M., Kennedy, J. F., Jiang, Y., Tang, J., & Liu, J. Chitosan and oligochitosan enhance ginger (Zingiber officinale Roscoe) resistance to rhizome rot caused by Fusarium oxysporum in storage. Carbohydrate Polymers, 2016, 151, 474-479.DOI:10.1016/j.carbpol.2016.05.103.

Q3:Completely describe methods properly (see my suggestions).

Rs: Thanks. According to your suggestions, we have made major changes to the materials and methods section of this paper.

Q4:Data presentation is ok but needs to re-arrange - see my comments and suggestions.

Rs: Thanks. We have reordered the discussion in the order in which the pictures of the experimental results appear.

Q5:Conclusion should be an over results of the study based on the target objectives.

Rs: Thanks. “This study aimed to 1) study the effects of eugenol on the reproductive and vegetative growth of F. oxysporum FOX-1 in vitro, 2) explore the mechanism of action of eugenol using microscopic observations and cell membrane integrity detection, and 3) evaluate the inhibitory efficacy of eugenol against Fusarium wilt in ginger seedlings. Our results will help to provide an efficient alternative approach to control Fusarium wilt.”

Based on the above objectives, we changed the conclusions of this paper.

“5. Conclusions

In this study, the soil-borne pathogen F. oxysporum FOX-1 was the target, and it was found that eugenol inhibited the growth of F. oxysporum FOX-1 mycelium and spore germination, and its inhibitory effect had a concentration-dependent characteristic. Further study of its mechanism of action from the perspective of the cell membrane found that eugenol acted on the cell membrane of Fusarium spinosum, damaged the cell structure of the pathogenic fungi, and destroyed the integrity of the cell membrane and permeability. Owing to the intracellular material spillover, the fungi cannot maintain normal physiological activities to achieve an antimicrobial effect. Finally, through potting experiments on bamboo root ginger for prevention and efficacy measurement, it was found that there was no significant difference between the control effect of treatment with 2 g/L eugenol and 8 g/L chlorothalonil. The results of this study can be used to provide a theoretical basis and promote the development of highly efficient, plant-derived, and environmentally friendly agents for the control of soil-borne diseases caused by Fusarium species.”

Q6:Recast the discussions following the sequence of presentation of data.

Rs: Thanks. We have adjusted the order of discussion based on the order in which the data appeared. More importantly, we have recursively discussed the entire experiment primarily based on the order of the objectives.

Q7:-provide an empirical data such as radial growth to justify figure1A. Percent germination rate in Fig 1B is ok.

Rs: Thanks. Table 1 describes the colony diameter data in Figure 1A.

Q8:Data are included in KV-micrographs and magnifications.

Rs: Thanks. We have added relevant information in the legend. SEM high voltage: 20.0 kv; SEM magnetic: 5.00 kv”

Reviewer 2 Report

The manuscript describes an interesting study aimed at investigating the effect of eugenol on Fusarium wilt. In vitro and in vivo studies are included. The methods are adequate, the English is good but the manuscript needs some adjustments to improve.

If the text does not indicate which Fusarium strain was used, it would be saying that, by indicating only the genus and species, all the strains of that genus and species give identical results. The identification of the strain (QY-4) must be indicated every time its use is discussed; not in the title but in the abstract, subtitles, tables, titles of figures and text in general. It should be written "F. oxysporum QY-4"

3.Results: it must be explained why ethanol is used in the tests. It is not clear

- Table 1: is it cited in the text? Its content should be cited and discussed

- 3.3, 3rd line: use of italics for genus and species

- Table 2: it should be located after being cited in the text, not before

Author Response

Responses to reviewer 2

Q1:If the text does not indicate which Fusarium strain was used, it would be saying that, by indicating only the genus and species, all the strains of that genus and species give identical results. The identification of the strain (Fox-1) must be indicated every time its use is discussed; not in the title but in the abstract, subtitles, tables, titles of figures and text in general. It should be written "F. oxysporum FOX-1"

Rs: Thanks. We have made a full text change to change “F. oxysporumin” to “F. oxysporum FOX-1.

Q2:Results: it must be explained why ethanol is used in the tests. It is not clear.

Rs: Thanks. “An amount of 3.752 mL of eugenol was aspirated and dissolved with 30% ethanol (v/v) to configure 4 g/L of eugenol reserve solution.” We specified 30% ethanol (v/v) as a co-solvent of eugenol in our method, so in order “to eliminate the effect of ethanol, we studied the effect of 30 % ethanol on the colony growth of F. oxysporum FOX-1 and found that it had little effect on the colony growth (p > 0.05; Figure 1A)”.

Q3:Table 1: is it cited in the text? Its content should be cited and discussed

Rs: Thanks. The contents of table1 have been referenced. In order to avoid duplication of the text with the content of the chart, we have quoted some data.

“The colony diameter of CK was (5.97 ± 012) cm after 5 d of incubation, and the colony diameters of eugenol treatments ranged from (0.60±0.00) to (4.95±0.25) cm (Table 1)”.

Q4:3.3, 3rd line: use of italics for genus and species

Rs: Thanks. We checked and revised the full text.

Q5:Table 2: it should be located after being cited in the text, not before

Rs: Thanks. We checked and revised the full text.

Reviewer 3 Report

Dear authors, your research paper presented to me for review is very interesting. Your scientific research is part of the current trend of using natural substances in plant protection products. The work is very valuable and I recommend it for publication in its present form. I only have reservations about the conclusions of the work, which are quite short. They do not fully address the three objectives set. Especially for the third objective "explore the mechanism of action of eugenol using microscopic observations and cell mem brane integrity detection" is not referenced in the results so my suggestion is to improve it.

Author Response

Q1:Dear authors, your research paper presented to me for review is very interesting. Your scientific research is part of the current trend of using natural substances in plant protection products. The work is very valuable and I recommend it for publication in its present form. I only have reservations about the conclusions of the work, which are quite short. They do not fully address the three objectives set. Especially for the third objective "explore the mechanism of action of eugenol using microscopic observations and cell mem brane integrity detection" is not referenced in the results so my suggestion is to improve it.

Rs: Thanks. “This study aimed to 1) study the effects of eugenol on the reproductive and vegetative growth of F. oxysporum FOX-1 in vitro, 2) explore the mechanism of action of eugenol using microscopic observations and cell membrane integrity detection, and 3) evaluate the inhibitory efficacy of eugenol against Fusarium wilt in ginger seedlings. Our results will help to provide an efficient alternative approach to control Fusarium wilt.”

Based on the above objectives, we changed the conclusions of this paper.

“5. Conclusions

In this study, the soil-borne pathogen Fusarium oxysporum FOX-1 was targeted, and it was found that eugenol inhibited the mycelium growth and spore germination of Fusarium oxysporum FOX-1, and the inhibitory effect was concentration-dependent. Further research on its mechanism of action from the perspective of the cell membrane found that eugenol acts on the cell membrane of Fusarium acanthium, destroying the cell structure of the pathogenic fungus, and destroying the integrity and permeability of the cell membrane. Due to the overflow of intracellular substances, fungi cannot maintain normal physiological activities to achieve antibacterial effects. Finally, through the bamboo root ginger pot experiment for prevention and curative effect determination, it was found that there was no significant difference between the control effects of 2 g/L eugenol and 8 g/L chlorothalonil. The results of this study can provide a theoretical basis for the control of soil-borne diseases caused by Fusarium, and promote the development of highly efficient, plant-derived, and environmentally friendly agents.

Round 2

Reviewer 2 Report

The authors have responded satisfactorily to the observations made and the manuscript has gained in quality. In my opinion, it deserves to be published.